# De Novo Assembly and Annotation of the Vaginal Metatranscriptome Associated with Bacterial Vaginosis

**DOI:** 10.3390/ijms23031621

**Published:** 2022-01-30

**Authors:** Won Kyong Cho, Yeonhwa Jo, Seri Jeong

**Affiliations:** 1College of Biotechnology and Bioengineering, Sungkyunkwan University, Seoburo 2066, Suwon 16419, Gyeonggi, Korea; wonkyong@gmail.com; 2Institute of Biotechnology and Bioengineering, Sungkyunkwan University, Seoburo 2066, Suwon 16419, Gyeonggi, Korea; yeonhwajo@gmail.com; 3Department of Laboratory Medicine, Kangnam Sacred Heart Hospital, Hallym University College of Medicine, 1 Singil-ro, Yeongdeungpo-gu, Seoul 07441, Korea

**Keywords:** vagina, transcriptome, de novo assembly, microbiome

## Abstract

The vaginal microbiome plays an important role in women’s health and disease. Here we reanalyzed 40 vaginal transcriptomes from a previous study of de novo assembly (metaT-Assembly) followed by functional annotation. We identified 286,293 contigs and further assigned them to 25 phyla, 209 genera, and 339 species. *Lactobacillus iners* and *Lactobacillus crispatus* dominated the microbiome of non-bacterial vaginosis (BV) samples, while a complex of microbiota was identified from BV-associated samples. The metaT-Assembly identified a higher number of bacterial species than the 16S rRNA amplicon and metaT-Kraken methods. However, metaT-Assembly and metaT-Kraken exhibited similar major bacterial composition at the species level. Binning of metatranscriptome data resulted in 176 bins from major known bacteria and several unidentified bacteria in the vagina. Functional analyses based on Clusters of Orthologous Genes (COGs) and Kyoto Encyclopedia of Genes and Genomes (KEGG) pathways suggested that a higher number of transcripts were expressed by the microbiome complex in the BV-associated samples than in non-BV-associated samples. The KEGG pathway analysis with an individual bacterial genome identified specific functions of the identified bacterial genome. Taken together, we demonstrated that the metaT-Assembly approach is an efficient tool to understand the dynamic microbial communities and their functional roles associated with the human vagina.

## 1. Introduction

The word “microbiome” is composed of two words, “micro” and “biome”, and refers to the collection of genetic material from all kinds of living microbiota, such as bacteria, archaea, fungi, and protozoa [1,2]. In general, phages, viruses, plasmids, prions, viroids, and free DNA do not belong to the microbiota because they are not living microorganisms [3]. The microbiota colonizes all kinds of living organisms, including humans, animals, and plants. In addition, microbiomes are ubiquitous; therefore, they can be found in most environments on earth, like soil and the sea [4]. The human microbiome is composed of all microorganisms that are present on or within the human tissues, such as the gastrointestinal tract, skin, oral mucosa, and vagina [5]. 

The vagina is the female genital tract extending from the vestibule to the cervix [6]. The vaginal microbiome is formed by the diverse microorganisms colonizing the human vagina [7]. Numerous studies have found that the microbiome in the vagina plays important roles in the health and disease of women [7,8,9]. The most common bacteria identified from healthy women are members of the genus *Lactobacillus*, including *Lactobacillus iners*, *Lactobacillus crispatus*, *Lactobacillus jensenii*, and *Lactobacillus gasseri* [10,11]. Bacterial vaginosis (BV) results from the overgrowth of bacteria residing in the vagina, causing vaginal inflammation [12]. A complex of several anaerobic bacteria, such as *Gardnerella*, *Atopobium*, *Dialister*, and *Peptoniphilus,* is known to be associated with BV [13]. Those pathogenic microbiota associated with BV have been linked to other conditions, such as sexually transmitted infections, pelvic inflammatory disease, and preterm birth [14]. 

The development of polymerase chain reaction, cloning, and sequencing techniques has enabled the examination of microbial communities. Furthermore, the introduction of phylogenetic genetic markers, such as 16S and 18S rRNA, as well as internal transcribed spacers (ITS), has promoted microbiome-associated studies [15,16]. In particular, the cultivation-independent approach using the 16S rRNA marker has been widely used for microbiome studies due to its straightforward nature and cost-effectiveness. Furthermore, the rapid development of next-generation sequencing techniques has facilitated microbiome studies based on the 16S rRNA marker. The 16S rRNA amplicon approach efficiently determines the community of microorganisms and their abundance in each sample with a small amount of DNA. However, the approach using the 16S rRNA marker cannot be applied to non-bacteria, such as Archaea and Eukarya. Moreover, the 16S rRNA marker does not provide the functional role of identified microorganisms. Furthermore, the results of 16S rRNA gene sequencing can vary depending on the choice of primers binding to the nine variable 16S subregions, suggesting the usefulness of the full 16S gene for better taxonomic resolution for species- and strain-level microbiome analysis [17,18]. 

To overcome the weakness of the 16S rRNA-based approach, DNA shotgun metagenomics, which sequences all genetic material from the sample, has been increasingly used. DNA shotgun metagenomics requires a greater amount of DNA material and is more costly than the 16S rRNA method. However, DNA shotgun metagenomics provides information about many genes from diverse microorganisms (not limited to bacteria). A recent study conducted a comparative analysis between 16S rRNA amplicon and DNA shotgun metagenomics with pediatric gut microbiome data [19]. This study showed that more bacteria genera were identified by 16S rRNA amplicon than DNA shotgun metagenomics. However, several bacterial genera were missed or underrepresented by the individual methods [19]. 

Many vaginal microbiome-associated studies have been conducted based on the 16S rRNA amplicon method. For example, several vaginal microbiomes for asymptomatic North American women from four different ethnic groups [7], women with preterm birth [20,21], pregnant women [22], women with atrophic vaginitis [23], and women with BV [24] have been investigated. By contrast, few studies have been conducted on the vaginal microbiome using DNA shotgun metagenomics [21,25]. 

In addition to DNA, RNA can also be good genetic material for microbiome studies. DNA-based metagenomics enables individual microbial genomes to be revealed from complex microbiomes, whereas RNA-based metatranscriptomics reveals the gene expression and functional roles of complex microbiomes. Thus, metatranscriptomics is a very useful approach for functional microbiome studies. However, it can be difficult to obtain enough RNA material from specific conditions, such as vaginal fluid (discharge). By contrast, the DNA material for 16S rRNA sequencing and DNA shotgun sequencing can be easily obtained using a swab. Therefore, only few vaginal microbiome studies using metatranscriptomics have been carried out [26,27]. For example, a previous study performed comparative metatranscriptomics between healthy and dysbiosis-associated vaginal samples [27] and another study examined the effect of metronidazole on BV [26]. 

Both sequencing methods and computational analysis are important factors in metatranscriptome analyses [28]. Due to the high complexity of metatranscriptomic sequence data generated by high-throughput sequencing, short-read based taxonomical classifiers, such as GOTTCHA [29], Kraken [30], and MetaPhlan2 [31], are very often used for shotgun metagenomic and metatranscriptomic data. In general, the assembled reads generating longer contigs are more accurate than the short-read based taxonomical classifiers. Thus, some tools, such as Centrifuge [32] and Kraken2 [33], using the longer contigs for taxonomical classification, have been developed. Several metagenomic assemblers, such as MEGAHIT [34], IDBA-UD [35], and metaSPAdes [36], are currently available. The assembled reads known as transcripts in metatranscriptomics can be further subjected to functional annotation with known metatranscriptome pipelines, such as SqueezeMeta [37], IMP [38], and MOSCA [39]. 

In this study, we reanalyzed 40 vaginal transcriptomes from a previous study [26] by de novo assembly followed by functional annotation using the SqueezeMeta analysis pipeline [37]. We evaluated the difference between the short-read based classifier and de novo assembler-based approach for the microbiome community study. Moreover, functional analyses were conducted by mapping reads on the assembled contigs. In addition, we obtained 172 bins containing known and unknown bacterial genomes with functional classification of expressed genes. 

## 2. Results

### 2.1. Summary of RNA Sequencing Data Associated with BV for Transcriptome Analyses

In this study, we used 40 RNA sequencing datasets associated with BV from a previous study [26]. Samples were collected from 14 different subjects whose age ranged from 20 to 48 years old (Appendix A). The number of samples we collected at different time points for individual subjects ranged from one to five (Appendix A). For example, five samples were collected from the four subjects 04-001, 06-004, 08-006, and 13-019. All information associated with the subjects and BV symptoms was described in detail in the previous study [26]. Among the many examined factors, we only focused on the BV state, which was divided into the positive group (22 samples) and the negative group (18 samples). 

### 2.2. De Novo Assembly of Vaginal Transcriptome

In order to obtain the vaginal transcriptome by de novo assembly, we downloaded paired sequenced raw data (80 fastq files) under the project accession number (PRJEB21446) from the NCBI’s SRA database (Appendix A). After quality trimming and deletion of human host-associated sequences, we conducted de novo transcriptome assembly. After combining the 40 different transcriptomes from 40 different samples, we obtained a vaginal transcriptome containing a total of 286,293 contigs with a total length of 315,794,287 bp. The longest contig was 109,165 bp, while the shortest contig was 200 bp. The N50 of the assembled contigs was 1802 bp. 

We taxonomically classified the assembled contigs. As shown in Table 1, 276,994 contigs (96.8%) were assigned to four super kingdoms. A total of 25 phyla, 32 classes, 58 orders, 99 families, 209 genera, and 339 species were identified. However, 926 contigs were not assigned to any known taxonomy.

### 2.3. Identification of Ribosomal RNAs and Transfer RNAs from Vaginal Transcriptome

Using the Barrnap program, we identified 269 rRNA sequences and further divided them into mitochondrial 12S rRNA (7 contigs), 16S rRNA (81 contigs) from bacteria and mitochondria, bacterial 23S rRNA (126 contigs), eukaryotic 28S rRNA (2 contigs), and bacterial 5S rRNA (53 contigs) (Appendix A). 

Moreover, we identified 4335 tRNA sequences and classified them into 25 different kinds of tRNAs (Appendix A). The most abundant tRNA was tRNA-Leu (412 contigs) followed by tRNA-Met (354 contigs) and tRNA-Ser (353 contigs) (Appendix A). Of the identified tRNAs, 106 contigs were identified as transfer-messenger RNAs (tmRNAs) (Appendix A). 

### 2.4. Prediction of Open Reading Frames (ORFs)

We identified 471,380 protein-coding gene sequences using the Prodigal program, which predicts ORFs in given sequences (Appendix A). In addition, we identified 23,528 orphan gene sequences or orphan open reading frames (ORFans) genes in which no orthologous proteins can be detected in closely related species (Appendix A). 

### 2.5. Taxonomical Classification of Identified Microorganisms

The identified contigs were taxonomically analyzed. Based on accumulated contig size, most identified contigs were assigned to bacteria 99.97% (Appendix A). However, we also identified contigs assigned to archaea, eukaryota, and viruses (Appendix A). At the species level, some eukaryotic contigs were assigned to *Piliocolobus tephrosceles*, *Homo sapiens*, *Mandrillus leucophaeus*, *Rhizopus delemar*, *Pongo abelii*, *Eimeria acervulina*, *Rhizopus oryzae*, and *Plasmodium ovale*. The identified viral contigs belonged to the orders Caudovirales, Petitvirales, and Ortervirales. 

Subsequently, we obtained 337 bacterial species from 40 vaginal transcriptomes after removing other organisms (Appendix A). We mapped individual raw data on the identified contigs. The proportion of mapped reads in each sample ranged from 6.3% to 97.28%. Based on the mapped reads, we examined bacterial abundance at the phylum and species levels in the 40 different samples. At the phylum level, 10 different bacterial phyla were identified (Appendix A). Of them, Actinobacteria, Firmicutes, Bacteroidetes, and Fusobacteria were identified as major phyla. The composition of bacterial phyla differed among different samples collected from the same subject. 

Next, we examined the relative abundance of identified bacterial species according to BV state (Figure 1). In the 22 samples belonging to the positive BV group, the most abundant bacterial species was *Gardnerella vaginalis*, followed by *Prevotella timonensis*, *Sneathia sanguinegens*, *Tissierellia bacterium*, and *Prevotella bivia*. However, in the 18 negative BV samples, two lactobacillus species, *L. iners* and *L. crispatus*, were abundantly present. In particular, *L. crispatus* was abundantly present in the five samples 08_003b, 08_006_b, 08_006_c, 08_006_d, and 08_006_e. 

We compared the abundance of abundantly present bacterial species between the negative and positive groups according to BV state. For that, we calculated normalized read counts (logCPM). Of eight representative bacterial species, the logCPM values in the negative group for *G. vaginalis*, *P. timonensis*, *S. sanguinegens*, *P. bivia*, and *Coriobacteriales bacterium* were much higher than those in the positive group (Figure 2). By contrast, the logCPM values for *L. iners* and *L. crispatus* in the positive group were much higher than those in the negative group (Figure 2). 

### 2.6. Alpha and Beta Diversity of Identified Microorganisms in Different Subjects and According to BV State

We analyzed the alpha diversity of identified bacterial species among 14 different subjects using the inverse Simpson and Shannon method (Figure 3). Five samples were obtained from each of the four subjects: 04-001, 06-004, 08-006, and 13-019. Of them, the difference in alpha diversity among the five samples was very high for two subjects (04-001 and 06-004), while the difference in alpha diversity among the five samples was low for two subjects (08-006 and 13-019). Subject 05-012 showed the highest alpha diversity, whereas subject 13-019 displayed the lowest alpha diversity among the 16 subjects. Although there were great differences in the diversity of bacterial species among subjects, the *p*-values calculated by the Kruskal–Wallis rank sum test were higher than 0.05. 

Next, we compared the alpha diversity between the two groups according to BV state (Figure 4). The diversity of bacterial species in the negative group was much lower than that in the positive group, as shown by the inverse Simpson and Shannon indexes (Figure 4). To determine whether there was a significant difference in bacterial species diversity between the negative and positive groups, we performed two different statistical tests. Both the Wilcoxon rank sum exact test (*p*-value less than 0.05) and Welch two-sample *t*-test results (*p*-value less than 0.05) clearly confirmed the significant difference in the alpha diversity between the two groups. 

We compared the beta diversity of bacterial species between the two groups according to BV state using the Bray–Curtis distance matrix. As shown in the heatmap, the two groups were noticeably clustered (Figure 5A). The distribution of Bray–Curtis distance values among samples in the positive group was narrower than that in the negative group (Figure 5B). The statistical results (*p*-values less than 0.05) of the Wilcoxon rank sum test with continuity correction confirmed that there were differences between samples within the negative and positive groups. In addition, the bacterial composition between the two groups (*p*-values less than 0.05) was significantly different (Figure 5C). 

### 2.7. Clustering of Samples and Identification of Biomarkers

Principal component analysis (PCA) was conducted to examine the bacterial composition among 14 different subjects at the species level (Figure 6A). PCA showed two groups of 40 samples. Group A contained 22 samples, while Group B contained 18 samples. Samples from the same subjects, such as 08_006_b and 08_006_c, were closely related. However, sample 05_012_a was distantly related to 05_012_b although they were derived from the same subject (Figure 6A). Principal coordinate analysis (PCoA) was also carried out to distinguish the 40 samples according to BV state (Figure 6B). PCoA revealed three groups within the 40 samples. The positive group (Group B) contained 22 samples. However, the negative group was divided into two groups: Group A (13 samples) and Group B (five samples). The five samples in Group B were 08_003_b, 08_006_b, 08_006_c, 08_006_d, and 08_006_e. 

We identified biomarkers between the two groups according to BV state using two different machine learning algorithms (Figure 6C). Both logistic regression and random forest algorithms identified four bacterial species. Of them, *Streptococcus mutans* was commonly identified by both algorithms. *Enterococcus termitis*, *Bifidobacterium choloepi*, and *L.* sp. were identified by logistic regression, whereas *Anaerococcus lactolyticus*, *P. bivia*, and *Peptoniphilus vaginalis* were identified solely by random forest. 

### 2.8. Functional Annotation of Identified Contigs According to Clusters of Orthologous Genes (COGs) and Identification of Differentially Expressed COGs

We mapped the raw data on the assembled contigs. The proportion of mapped reads ranged from 6.3% (04_001_b) to 97.25% (13_022_a) (Appendix A). There were great differences in the proportion of mapped reads among different samples from the same subject. 

By performing a BLASTX search against the identified COGs, we identified a total of 7277 COGs. Based on the mapped reads of the assigned COGs, we identified differentially abundant functions by comparing the positive group to the negative group according to BV state. Based on the twofold changes and adjusted *p*-values less than 0.01, 4782 differentially abundant functions (65.7%) between the two groups were identified (Figure 7A and Appendix A). Hierarchical clustering analysis (HCL) revealed two groups of samples: negative (18 samples) and positive (22 samples) (Figure 7B). Moreover, the COGs were divided into upregulated (4043 COGs) and downregulated (739 COGs) by comparing the positive group to the negative group according to BV state (Appendix A). The six representative COGs showed differential abundance between the two groups (Figure 8). 

The abundance of identified COGs was very different between the two groups (Figure 9). For example, COG0266, COG0344, and COG1922 were highly upregulated in the negative group, while COG0250, COG1882, and COG1480 were highly downregulated in the negative group. 

COGs can be further divided into 27 functional categories. The number of up- and downregulated COGs in the 27 functional categories was analyzed (Appendix A). Unknown function was dominant in the upregulated (60.1%) and downregulated groups (47.8%). The number of upregulated COGs was 5.5 times higher than that of downregulated COGs. In general, the number of COGs in the 27 functional categories in the upregulated group was much higher than that in the downregulated group. For instance, the number of COGs associated with functional categories, such as energy production and conversion (139 COGs) and DNA replication, recombination, and repair (35 COGs), in the upregulated group was more than 10 times higher than that in the downregulated group. In addition, three functional categories (cell cycle control, cell division, and chromosome partitioning) (7 COGs), cell motility (13 COGs), coenzyme transport and metabolism (8 COGs), and lipid transport and metabolism (3 COGs)) were only identified in the upregulated group. 

We calculated the relative abundance of COGs in the upregulated and downregulated groups (Figure 9 and Appendix A). As compared to the upregulated group, several functional categories (e.g., carbohydrate transport and metabolism (6.5%), cell envelope biogenesis, outer membrane (3.1%), defense mechanisms (4.7%), general function prediction only (8.3%), transcription (4.9%), translation, ribosomal structure, and biogenesis (2.3%), and unclassified (3.9%)) were higher in the downregulated group. 

### 2.9. Functional Annotation of Identified Contigs According to Kyoto Encyclopedia of Genes and Genomes (KEGG) Pathways and Identification of Differentially Expressed Enzymes

We identified a total of 3501 enzymes from 40 vaginal transcriptomes (Appendix A). Again, we identified differentially abundant enzymes by comparing the BV-positive group to the BV-negative group based on the number of mapped reads using DESeq2. Of the 3501 enzymes, 2433 enzymes were differentially regulated between the two groups. The identified 2433 enzymes were assigned to 315 KEGG pathways. Of the 315 KEGG pathways, 122 pathways were commonly identified in both the up- and downregulated groups, while 174 and 19 enzymes were specific to the upregulated and downregulated groups, respectively. The most abundant pathways were the metabolic pathways (map01100) followed by the biosynthesis of secondary metabolites (map01110) and microbial metabolism in diverse environments (map01120). 

The number of identified enzymes in the upregulated KEGG group was much higher than that in the downregulated KEGG group (Appendix A). For example, many primary and secondary metabolisms, such as carbon metabolism (5 times), porphyrin metabolism (14.3 times), glycine, serine, and threonine metabolism (11 times), glyoxylate and dicarboxylate metabolism (33 times), and propanoate metabolism (16.5 times), were enriched in the upregulated group. We calculated the relative abundance of identified enzymes in each group according to 28 representative KEGG pathways (Figure 10). The up- and downregulated groups exhibited similar proportions of enzymes in each pathway (Figure 10). However, we found that flagellar assembly (31 enzymes), bacterial chemotaxis (17 enzymes), biofilm formation for *Vibrio cholerae* (16 enzymes), *Escherichia coli* (14 enzymes), and *Pseudomonas aeruginosa* (11 enzymes) were highly enriched in the upregulated group. The involvement of identified enzymes in the three pathways (flagellar assembly, bacterial chemotaxis, and biofilm formation for *V. cholerae*) is depicted in Figure 11. 

### 2.10. Binning of Metatranscriptome Data and Functional Classification of Identified Bins

Using MetaBAT2, we conducted metatranscriptome binning to obtain single genomes from the 40 vaginal transcriptome datasets, which resulted in 176 assembled bins (Figure 12 and Appendix A). The completeness of the 176 bins ranged from 0% (Bin144) to 92.5% (Bin166). The genome size of the 176 bins ranged from 106,626 bp (Bin005) to 4469,455 bp (Bin104). The GC content of the 176 bins ranged from 27.8% (Bin042) to 54.2% (Bin134). 

The 176 bins were taxonomically assigned to four phyla (162 bins), eight classes (154 bins), nine orders (137 bins), nine families (124 bins), 10 genera (114 bins), and 13 species (101 bins) (Appendix A). Many bins were not assigned to any taxonomy. For example, 14 bins at the phylum level, 22 bins at the class level, 39 bins at the order level, 62 bins at the genus level, and 101 bins at the species level were not taxonomically assigned. 

Raw data from the 40 different samples were again mapped on the sequences of the 176 bins (Appendix A). Based on the number of mapped reads, we examined the relative abundance of identified bacteria at the genus level (Appendix A). In the BV-negative group, the genus *Lactobacillus* was dominant, whereas some samples contained several unassigned bins. By contrast, several bacterial genera, such as *Gardnerella*, *Sneathia*, *Prevotella*, and *Peptoniphilus*, as well as many unassigned bins, were dominant in the BV-positive group. At the species level, the BV-negative group contained mostly *L. iners* and *L. crispatus* (Appendix A). Interestingly, *L. crispatus* was dominantly present only in the five samples 08_003b, 08_006_b, 08_006_c, 08_006_d, and 08_006_e. In the BV-positive group, unassigned bins were dominantly present in most samples. In addition, other bacterial species, such as *G. vaginalis*, *L. iners*, *S. sanguinegens*, and *P. timonensis*, were present together.

Mapping results on the 176 bins identified 10 major genera in the vaginal transcriptome samples: *Lactobacillus*, *Sneathia*, *Megasphaera*, *Prevotella*, *Gardnerella*, *Mobiluncus*, *Aerococcus*, *Peptostreptococcus*, *Fannyhessea*, and *Peptoniphilus* (Figure 13A). All genera except *Lactobacillus*, which was abundantly present in the negative group, were preferentially present in the positive group. At the species level, 13 bacterial species were major species in the vaginal transcriptome. Three bacterial species, *L. iners*, *L. crispatus*, and *L. jensenii*, were dominantly present in the negative group, while the other 10 bacterial species were abundantly present in the positive group (Figure 13B). 

We conducted PCoA analyses using the mapped reads on the 176 bins (Appendix A). We identified three groups of the 40 samples (Figure 14). Group A and Group B belonged to the BV-negative group, while Group C belonged to the BV-positive group. Group A contained five samples in which *L. crispatus* was abundantly present, while Group B contained 13 samples in which *L. iners* was abundantly present. Group C included 22 samples in which 10 major bacterial species, such as *G. vaginalis*, *P. timonensis*, and *Mobiluncus mulieris*, were abundantly present.

We calculated the number of enzymes assigned to the 97 KEGG pathways for nine major bacterial species. Many metabolism-related pathways, such as ABC transporters, ribosomal, purine metabolism, and protein export, were commonly identified in the nine bacterial species (Figure 15). However, some KEGG pathways were specifically present in a certain bacterial species. For example, lipopolysaccharide biosynthesis, fatty acid metabolism, histidine metabolism, phenylpropanoid biosynthesis, sphingolipid metabolism, reductive carboxylate cycle, propanoate metabolism, carbon fixation in photosynthetic organisms, biotin metabolism, benzoate degradation via CoA ligation, beta-Alanine metabolism, and phenylalanine metabolism were only identified from *P. timonensis*. Flagellar assembly, trinitrotoluene degradation, ether lipid metabolism, and bacterial chemotaxis were identified from *M. mulieris*. Benzoate degradation via hydroxylation was identified from *L. crispatus*. D-arginine and D-ornithine metabolism were identified from *Coriobacteriales bacterium*, while novobiocin biosynthesis was identified from *Aerococcus christensenii*.

### 2.11. Comparison of Three Different Approaches for Identification of Microorganisms from Vagina Samples

We compared the number of identified bacterial species among three different approaches from the same research group (Appendix A). The first study used 16S rRNA amplicon sequencing from 96 vaginal samples [24]. The second study used metatranscriptomics of 40 vaginal samples using the Kraken program by mapping raw sequence reads on the reference bacterial genomes (metaT-Kraken) [26]. The third study also used the same dataset from the 40 vaginal samples, but the microorganisms were identified by de novo transcriptome assembly followed by a BLASTX search against a non-redundant (NR) protein database (metaT-Assembly). From 16S rRNA amplicon, 133 bacterial species were identified (Appendix A). Although the same dataset was used, 103 bacterial species were identified from metaT-Kraken, whereas 337 bacterial species were identified from the metaT-Assembly approach. From the three different studies, 34 bacterial species that were major bacteria in the vagina were commonly identified (Appendix A).

## 3. Discussion

As suggested in many previous studies, different experimental approaches, such as sequencing and analytical methods, can affect the results of microbiome studies even if the same materials are used [40]. Here, we used 40 different vaginal transcriptome datasets from a previous study and conducted de novo transcriptome assembly followed by annotation of identified microorganisms. Our results demonstrated that de novo assembly of transcriptome data for the microbiome study was very efficient for detailed taxonomical classification and estimation of abundance of a complex vaginal microbiota.

It has been generally regarded that the approach based on 16S rRNA amplicon could determine the microbiota composition of given samples better than shotgun sequencing-based approaches, such as metagenomics or metatranscriptomics. Our comparative analyses among three different microbiome approaches revealed that there was no significant difference in the number of identified bacteria at the species level between 16S rRNA amplicon and metaT-Kraken. However, the number of bacterial species identified by metaT-Assembly was much higher than that by 16S rRNA amplicon (2.53 times) and metaT-Kraken (3.27 times). There are several possible reasons why metaT-Assembly revealed the highest number of bacterial species among the three different approaches. The first possible reason is the database used for the identification of microorganisms. For metaT-Kraken, a database containing all available complete bacterial genome sequences was used, whereas an NR protein database was used for metaT-Assembly. An NR protein database covers most known organisms, while the other two methods only include a limited number of bacteria for which genome sequences are available. Furthermore, the approach based on BLASTX search using translated nucleotide sequences against NR sometimes leads to false positive taxonomy identification, although BLASTX search has very high gene prediction accuracy [41]. For example, complete genome sequences of an organism obtained using BLASTX search can be matched to many organisms that are closely related to the target organism. Regardless of the analytical methods, the proportion of major bacteria in each sample was very similar between metaT-Kraken and metaT-Assembly. This result suggests that both methods are very valuable for microbiome studies using transcriptome data. The main advantage of metaT-Kraken might be the speed of data analysis as compared to metaT-Assembly, which requires a lot of time for de novo assembly, followed by mapping and high-performance computation [30]. A previous study evaluated the performance of 20 metagenomic programs using experimental and simulated datasets in detail [42]. As a result, the choice of data analysis method could vary depending on the purpose of the study, cost, and time.

The lack of bacterial genomes in the reference database could result in the misinterpretation of microbiome analyses. Therefore, it is very important to obtain as many reference bacterial genomes as possible for microbiome studies. The most common method to obtain complete microbial genomes might be sequencing of culturable microorganisms from the target tissues or conditions. For instance, a recent study sequenced the complete genome of 1520 culturable bacteria that were collected from more than 6000 bacteria present in the fecal samples of healthy humans [43]. The second method is the direct sequencing of both culturable and unculturable bacteria using next-generation sequencing followed by metagenomic binning and assembly [44,45]. Here, we obtained 176 metatranscriptome-assembled genomes (MAGs). As the MAGs were derived from the transcriptomes in this study, the completeness of MAGs was lower than that of metagenome-assembled genomes [25]. However, the assembled bacterial transcriptome in this study provided the most important genes: those that were highly expressed and associated with the human vagina and BV. At the species level, the frequently identified MAGs in the vagina were *G. vaginalis*, *L. iners*, *P. timonensis*, *S. sanguinegens*, and *T. bacterium*, and their abundance was also high in the 40 vagina samples. This result indicated that the abundance of bacterial species in the metatranscriptomes was highly correlated with the number of MAGs. At the kingdom level, 14 MAGs were not matched to any known bacteria. Five *Actinobacteria-* and three *Firmicutes*-associated MAGs were identified as novel bacteria at the phylum level. At the genus level, 17 MAGs assigned to *Gardnerella*, two MAGs assigned to *Lactobacillus*, nine MAGs assigned to *Prevotella*, and 12 MAGs assigned to *Sneathia* were identified; however, their species levels were not determined. These results suggest that there could be many unidentified bacterial species from the human vagina. In fact, there are still many unidentified and unculturable bacterial species in the human vagina. A previous study reported BV-associated bacterium 1 (BVAB1) from the human vagina [46]. The complete genome of BVAB1, named *Candidatus* Lachnocurva vaginae, was recently obtained using two different sequencing techniques, PacBio Sequel II and HiSeq platforms, followed by de novo assembly [46].

As previously reported, two *Lactobacillus* species, *L. iners* and *L. crispatus*, dominated the microbiome of the 18 non-BV samples. Interestingly, there was only one single dominant *Lactobacillus* species, and none of non-BV samples shared two different *Lactobacillus* species. This result is consistent with the many previous results indicating that the vaginas of healthy women are dominated by a single *Lactobacillus* species [47]. In contrast, 22 BV-associated samples contained a complex of microorganisms, although *G. vaginalis* was the dominant bacterial species in many BV-associated samples. Moreover, the proportion of *L. iners* was relatively high in several BV-associated samples, whereas the proportion of *L. crispatus* was relatively low in several BV-associated samples. As previously reported, *L. iners* coexisted with BV-associated bacteria, such as *G. vaginalis*, and did not perform a protective role against BV [48]. By contrast, human *L. crispatus* isolates are regarded as healthy probiotics used as therapeutic agents to treat dysbiosis [49].

One of the main advantages of metatranscriptomics could be to reveal functional roles of BV-associated microbiomes. Functional analyses based on COGs and KEGG pathways revealed that the microbiomes associated with BV expressed huge numbers of genes compared to non-BV samples. Differentially expressed gene (DEG) analysis using the number of mapped reads on COGs found that the number of upregulated COGs was five times higher than that of downregulated COGs. Therefore, it was very difficult for us to identify specific gene functions associated with BV in the upregulated COGs since the number of identified COGs for the upregulated group was much higher than that for the downregulated group in most functional categories. However, the approach of dividing the COGs into 27 functional categories narrowed down the specific functions enriched in the upregulated and downregulated COGs. For instance, some specific COG functional groups were only identified for the upregulated group, such as cell cycle control, cell division, chromosome portioning, cell motility, and coenzyme metabolism, while carbohydrate transport and metabolism, cell envelope biogenesis, and defense mechanisms were specifically identified from the downregulated group. Again, the number of DEGs in the upregulated group was found to be much higher than that in the downregulated group by KEGG pathway analysis. The active gene expression in the BV-associated samples suggests that the consortium of microbiota promoted the gene expression machinery to develop BV in the subject. Moreover, transcripts expressed by a dominant *Lactobacillus* species in the non-BV associated samples should be much lower than in the BV-associated samples. Simply, the large difference in the number of DEGs between BV and non-BV samples could be determined by the number of microorganisms actively living in the vagina.

Although COGs and KEGG pathways provided a comprehensive overview of the large number of functional groups for DEGs between the BV- and non-BV-associated samples, the identified functions were derived from the complex, not from an individual bacterial species. However, the approach based on the binning of MAGs followed by KEGG pathway analyses very efficiently revealed the functional roles of individual bacterial genomes in the complex vaginal microbiome. For example, we found that some functions associated with flagellar assembly and bacterial chemotaxis enriched in BV-associated samples were derived from *M. mulieris*. It has been known that *M. mulieris* is a curved and anaerobic bacteria whose cells are motile and have multiple subpolar flagella [50] and is often identified from the vagina [51]. Bacterial chemotaxis is defined as the direct movement of bacteria toward environmental conditions [52]. Our results suggest that the mobile ability of *M. mulieris* seems to be strongly connected with BV. Benzoate is used as a model compound to study the bacterial catabolism of aromatic compounds, which are the second most widely identified organic compounds in nature next to carbohydrates [53]. Benzoate degradation via hydroxylation, specifically identified from *L. crispatus*, indicates that anaerobic biodegradation of benzoate might relate to the beneficial effects of *L. crispatus* in the vaginas of healthy women.

KEGG analysis revealed that biofilm formation-associated genes for *V. cholerae*, *E. coli*, and *P. aeruginosa* were strongly upregulated in BV-associated samples. However, those three bacterial species were not identified in our study. It is likely that many orthologous genes associated with biofilm formation are upregulated in BV-associated samples. A recent study demonstrated that *G. vaginalis* along with other vaginal pathogenic bacteria could modulate the virulence of *G. vaginalis* and the formation of polymicrobial BV biofilms [54]. Of the frequently identified BV-associated bacteria in this study, many metabolic-associated genes for *P. timonensis* were specifically upregulated in the BV-associated samples. A recent study showed that *P. timonensis* is a strong inducer of inflammatory responses by inducing immune activation, which was demonstrated by using antigen-presenting dendritic cells [55].

## 4. Conclusions

In this study, reanalysis of 40 vaginal transcriptome datasets based on de novo assembly and functional annotation was useful to define the vaginal microbiota community associated with BV. We found that metaT-Assembly identified a higher number of bacterial species than 16S rRNA and metaT-Kraken; however, metaT-Assembly and metaT-Kraken exhibited similar major bacterial composition at the species level. Binning of metatranscriptome data provided single transcriptomes of major known bacteria as well as several unidentified bacteria in the vagina. Functional analyses based on COGs and KEGG pathways suggested that a high number of transcripts were expressed by the microbiome complex in the BV-associated samples as compared to that of the non-BV-associated samples. The KEGG pathway analysis with an individual MAG identified the specific functions of the identified individual bacterial genome. Taken together, we demonstrated that the metaT-Assembly approach is an efficient tool to understand the dynamic microbial communities and their functional roles associated with the human vagina.

## 5. Materials and Methods

### 5.1. Trimming, De Novo Transcriptome Assembly, and Deletion of Human Sequences

To delete human-associated sequences, raw fastq files were mapped on the human reference transcriptome data (GRCh38) using the BBDuk program with the following command (bbduk.sh in = reads.fq out = clean.fq qtrim = rl trimq = 20 minlen = 50) (https://jgi.doe.gov/data-and-tools/bbtools/bb-tools-user-guide/bbduk-guide/) (1 September 2021). During this procedure, we deleted raw sequences whose Phred score was less than 20 and whose sequence length was less than 50. After that, trimmed sequences were subjected to de novo transcriptome assembly using MEGAHIT, a very fast assembler frequently used in metagenomics assembly, with default parameters. De novo transcriptome assembly was conducted for an individual library, resulting in 40 different assembled transcriptomes. The 40 transcriptomes were all combined and subjected to a BLASTX search against an NR protein database using the DIAMOND program with E-value 1 × 10^−3^ as a cutoff [56]. The BLASTX results were analyzed by the MEGAN6 program for taxonomical classification [57]. Based on BLASTX and taxonomical classification, we deleted contigs associated with human hosts and contaminants. Finally, the 286,293 clean contigs were subjected to metatranscriptome analyses. Contig statistics were performed using PRINSEQ [58].

### 5.2. Identification and Classification of rRNA and tRNA

All metatranscriptome analyses were conducted using SqueezeMeta (version 1.4.0), which provides a fully automated pipeline for metagenomics/metatranscriptomics [37]. The assembled contigs were subjected to RNA prediction and classification. rRNA sequences were predicted by Prodigal v2.6.2 [59]. To detect the rRNA genes (5S, 16S, 23S), Barrnap was used (https://github.com/tseemann/barrnap) (1 September 2021). We used the ARAGORN program to identify tRNA and tmRNA genes [60]. The RDP classifier was used for the taxonomic classification of identified 16S rRNA sequences [61].

### 5.3. Prediction of ORFs, Taxonomical Classification, and Functional Annotation

The contigs were used for the prediction of ORFs by Prodigal. For taxonomical classification and functional annotation of identified contigs, all contigs were subjected to BLASTX search against an NR protein database (eggnog database) and KEGG database using DIAMOND with default parameters [62,63,64]. In addition, we conducted HMM homology searches against the Pfam database using HMMER3 [65,66]. We used the MinPath program [67] to identify enzymes assigned to KEGG pathway [64] and MetaCyc databases [68]. Taxonomy assignment was conducted using the SqueezeMeta program [37]. All microbiome-associated analyses—including the abundance analysis, alpha and beta diversity analyses, identification of biomarkers, and clustering (e.g., PCA and PCoA)—were performed using the animalcules tool implemented in R [69].

### 5.4. Mapping of Reads and Normalization of Mapped Read Number

All read mapping was conducted using Bowtie2 [70]. For example, raw sequence reads in each library were aligned on the contigs, COGs, and KEGG pathways. The obtained read numbers were also used for the calculation of reads per kilobase per million (RPKM) and transcripts per million (TPM).

### 5.5. Identification of Differentially Expressed COGs and Enzymes

The number of mapped reads on individual COGs and enzymes in KEGG pathways was subjected to the analysis of DEG using DESeq2 [71]. For simplicity, we compared the BV-associated group (22 samples) to the non BV-associated group (18 samples). As a result, we identified differentially expressed COGs and enzymes between the two groups based on the twofold changes and set *p*-values less than 0.01 as the cutoff.

### 5.6. Identification of Biomarkers at Species Level Using Logistic Regression and Random Forest Algorithms

We used two algorithms, logistic regression and random forest, to identify biomarkers at the species level between the two different conditions according to BV state. For that, we used the following parameters for the two algorithms: number of CV folds = 3; number of CV repeats = 3; top biomarker proportion = 0.01.

### 5.7. Reconstruction of Bacterial Genomes from Vaginal Metatranscriptome Dataset

All contigs were subjected to binning using two different programs, MaxBin2 [72] and MetaBAT [73]. The DAS tool was used for binning integration [74]. Finally, the quality of identified bins was assessed using CheckM [75]. The assembled bins were visualized by the VizBin program [76]. Taxonomic assignment and prediction of KEGG and MetaCyc pathways for individual bins were conducted using SqueezeMeta [37].

## Figures and Tables

**Figure 1 ijms-23-01621-f001:**
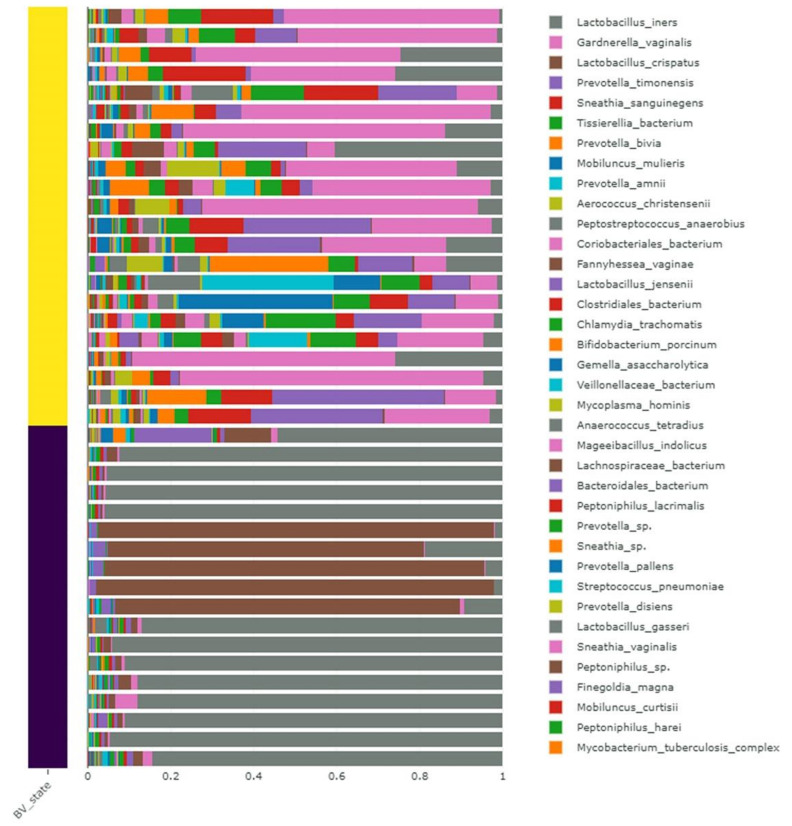
Relative abundance of identified bacteria in 40 different samples at species level. Yellow and purple bars indicate positive and negative samples for BV state, respectively. Only abundantly present bacterial species are listed in the legend.

**Figure 2 ijms-23-01621-f002:**
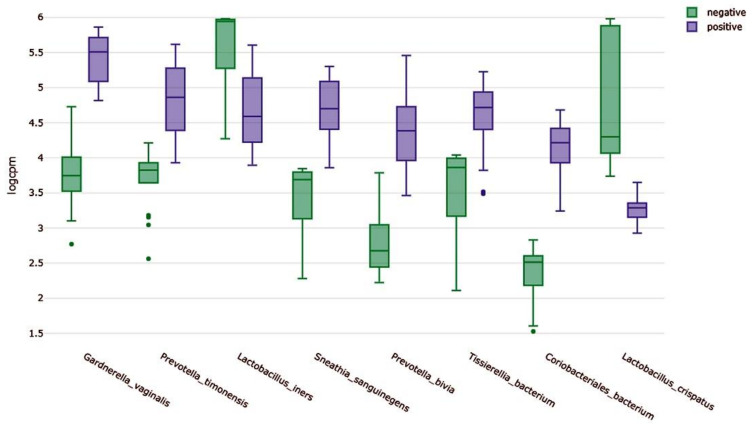
Bar charts displaying normalized read counts (logCPM) for eight representative bacterial species between negative and positive groups.

**Figure 3 ijms-23-01621-f003:**
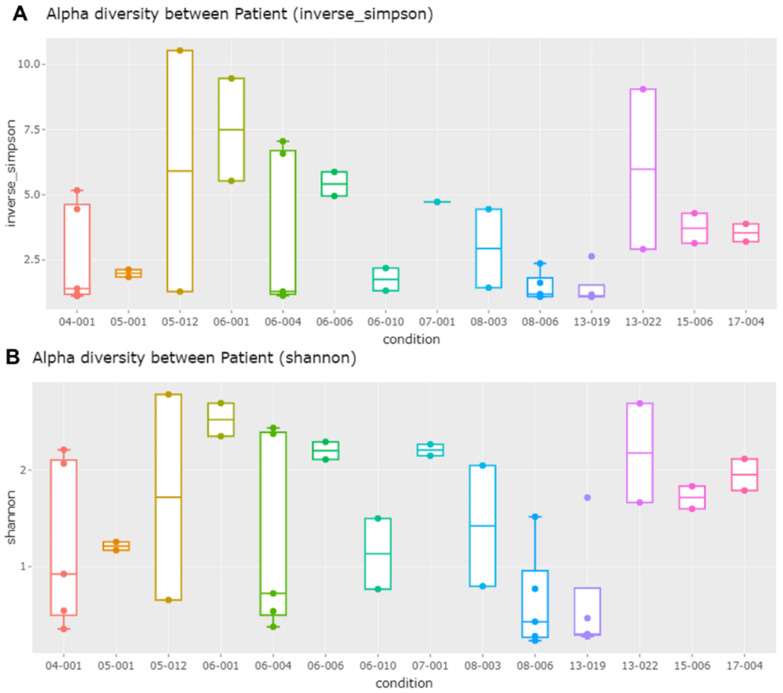
Comparison of alpha diversity of bacterial species among different subjects using inverse Simpson (**A**) and Shannon (**B**) methods. The Kruskal–Wallis rank sum test was conducted, resulting in *p*-values of 0.0897 and 0.09 for the inverse Simpson and Shannon, respectively.

**Figure 4 ijms-23-01621-f004:**
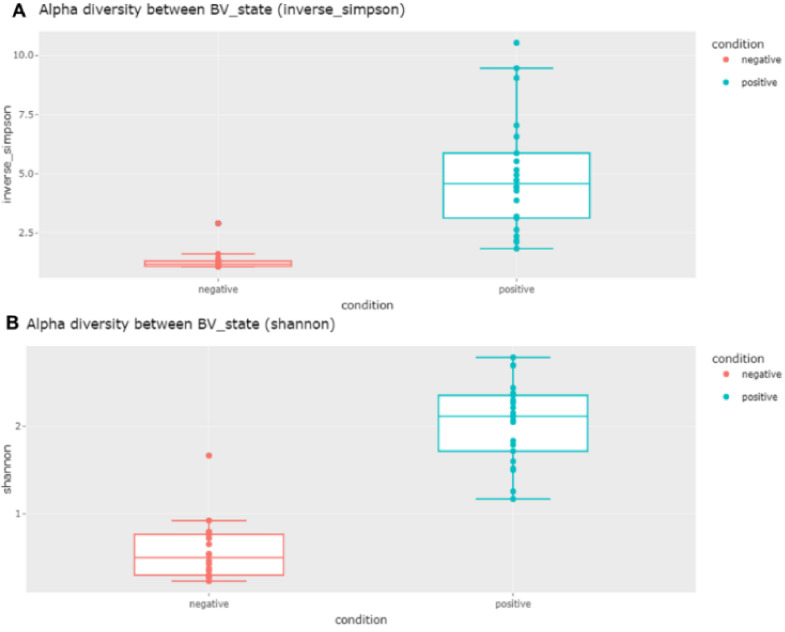
Comparison of alpha diversity of bacterial species between negative and positive groups using inverse Simpson (**A**) and Shannon (**B**) methods. The Wilcoxon rank sum exact test and Welch two-sample *t*-test were carried out to confirm the alpha diversity of bacterial species between the two groups: *p*-values of 3.35 × 10^−10^ (Wilcoxon) and 5.87 × 10^−7^ (Welch) were obtained by inverse Simpson, and *p*-values of 3.35 × 10^−10^ (Wilcoxon) and 4.64 × 10^−14^ (Welch) were obtained by Shannon.

**Figure 5 ijms-23-01621-f005:**
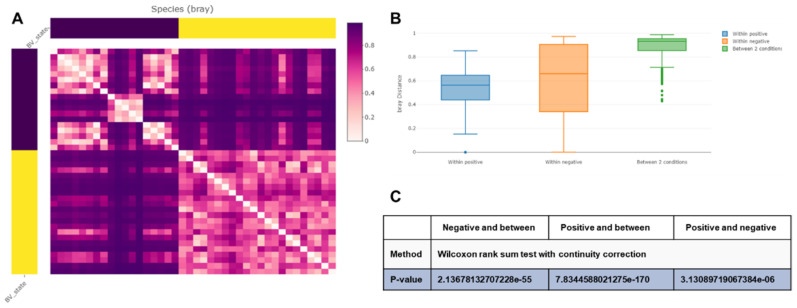
Beta diversity analyses of bacterial species according to bacterial vaginosis (BV) state using Bray–Curtis distance matrix. Heatmap using Bray–Curtis distance matrix showing two clustered groups of samples (**A**). Boxplot showing Bray–Curtis distance within samples and conditions (**B**). Results of Wilcoxon rank sum test with continuity correction with *p*-values (**C**).

**Figure 6 ijms-23-01621-f006:**
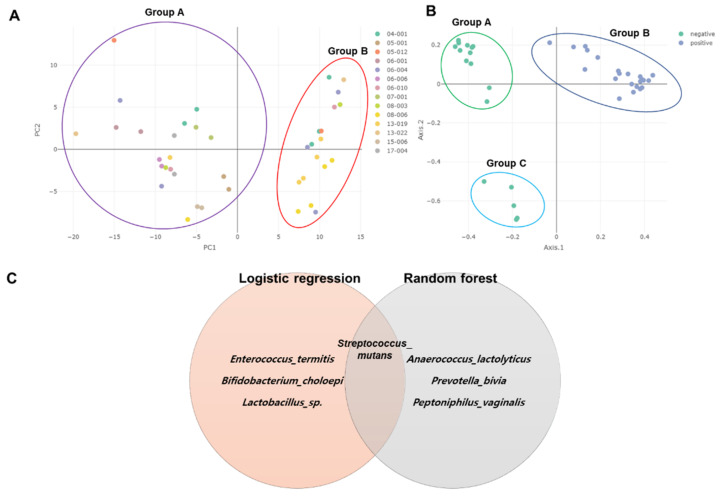
Dimension reduction analyses and identification of biomarkers. Principal component analysis (PCA) and principal coordinate analysis (PCoA) were conducted for 16 subjects (**A**) and two groups of BV state (**B**), respectively. Identification of biomarkers by two different machine learning algorithms (**C**).

**Figure 7 ijms-23-01621-f007:**
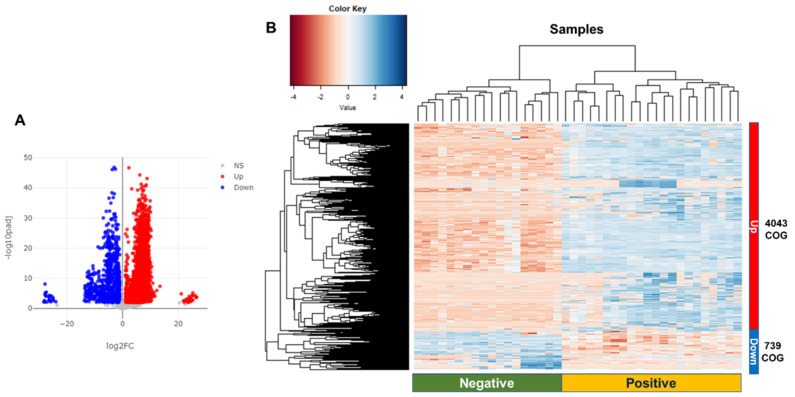
Identification of differentially abundant Clusters of Orthologous Genes (COGs) functions and hierarchical clustering analysis (HCL) of identified COGs. Volcano plots show distribution of fold changes and adjusted *p*-values for the differentially abundant functions identified by DESeq2 (**A**). Blue- and red-colored dots indicate the downregulated (Down) and upregulated (Up) COGs, respectively, by comparing the positive group to the negative group according to BV state. NS indicates no significant COG functions. Using the number of mapped reads on the assigned COGs, HCL was conducted (**B**).

**Figure 8 ijms-23-01621-f008:**
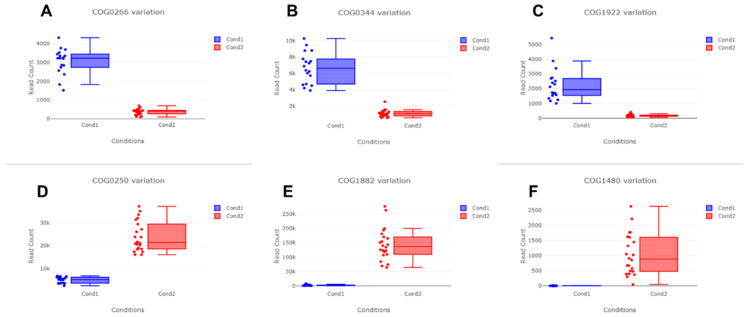
Six representative COGs identified as differentially abundant COGs between two groups. Blue (Cond1) and red (Cond2) colors indicate negative and positive groups, respectively. (**A**) COG0266 formamidopyrimidine-DNA glycosylase; (**B**) COG0344 phospholipid biosynthesis protein PlsY; (**C**) COG1922 UDP-N-acetyl-D-mannosaminuronic acid transferase; (**D**) COG0250 transcription antitermination factor NusG; (**E**) COG1882 pyruvate-formate lyase; and (**F**) COG1480 predicted membrane-associated HD superfamily hydrolase.

**Figure 9 ijms-23-01621-f009:**
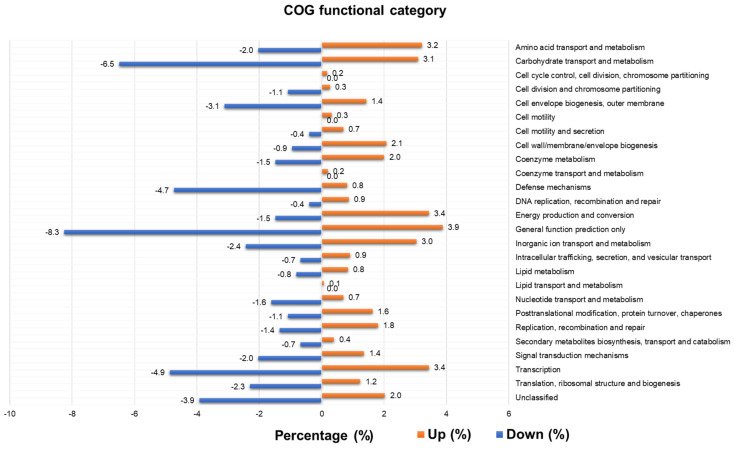
Relative abundance of identified COGs in upregulated and downregulated groups.

**Figure 10 ijms-23-01621-f010:**
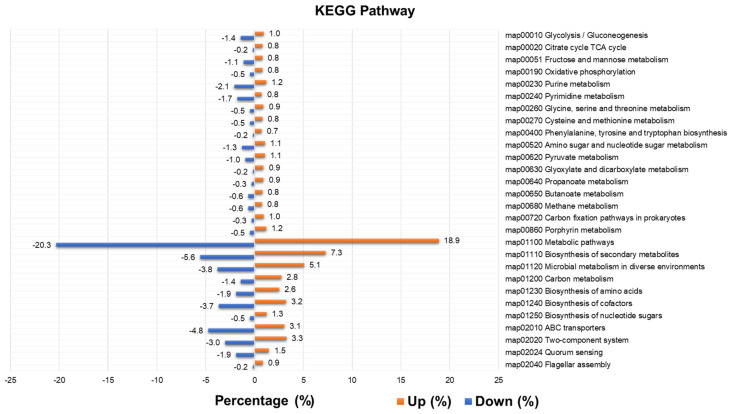
Relative enzyme abundance of 28 representative Kyoto Encyclopedia of Genes and Genomes (KEGG) pathways in upregulated and downregulated groups.

**Figure 11 ijms-23-01621-f011:**
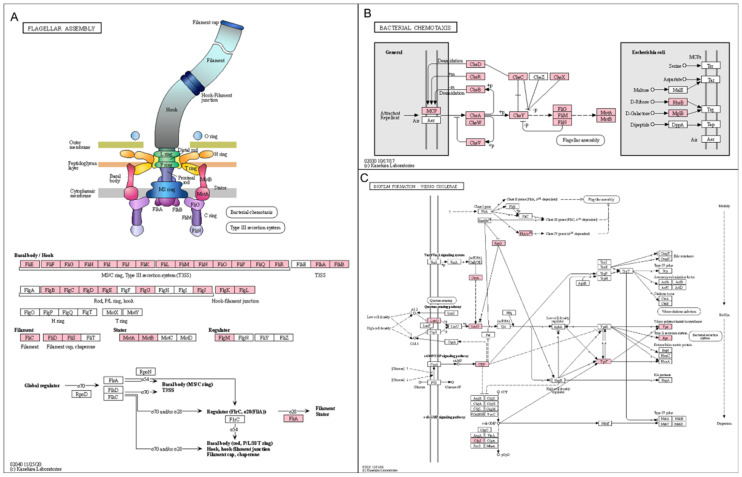
Three KEGG pathways enriched in upregulated group. Identified enzymes assigned to pathways associated with (**A**) flagellar assembly, (**B**) bacterial chemotaxis, and (**C**) biofilm formation for V. cholerae. Pink-colored boxes indicate enzymes identified from vaginal transcriptome.

**Figure 12 ijms-23-01621-f012:**
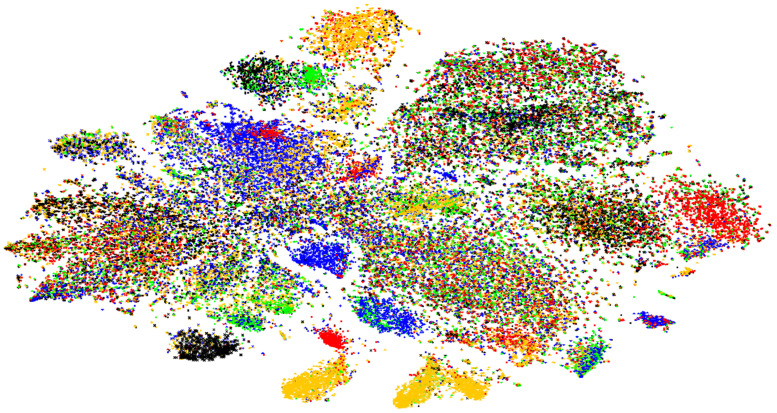
Visualization of 176 assembled metatranscriptome bins. Contigs assigned to 176 individual bins were grouped and visualized by the VizBin program. Each contig was labeled using different shapes and colors according to the 176 bins.

**Figure 13 ijms-23-01621-f013:**
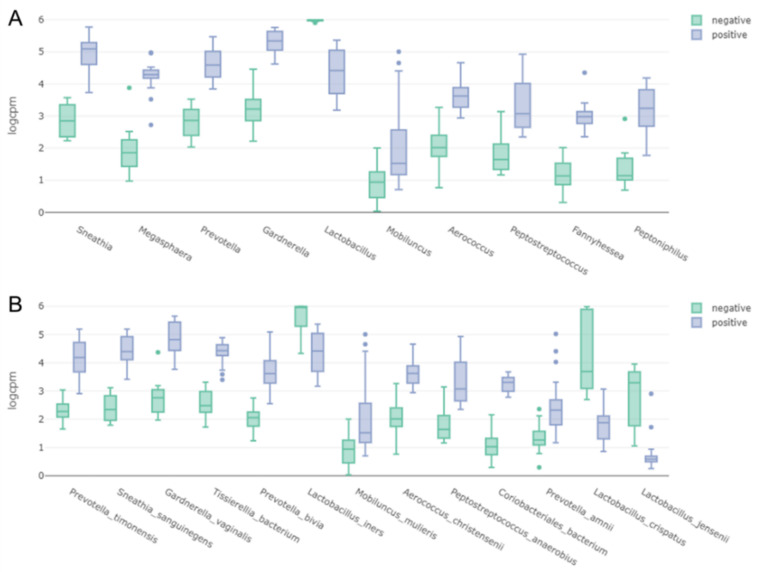
Bar charts displaying normalized read counts (logCPM) for 10 representative bacterial genera (**A**) and 13 species (**B**) between BV negative and positive groups.

**Figure 14 ijms-23-01621-f014:**
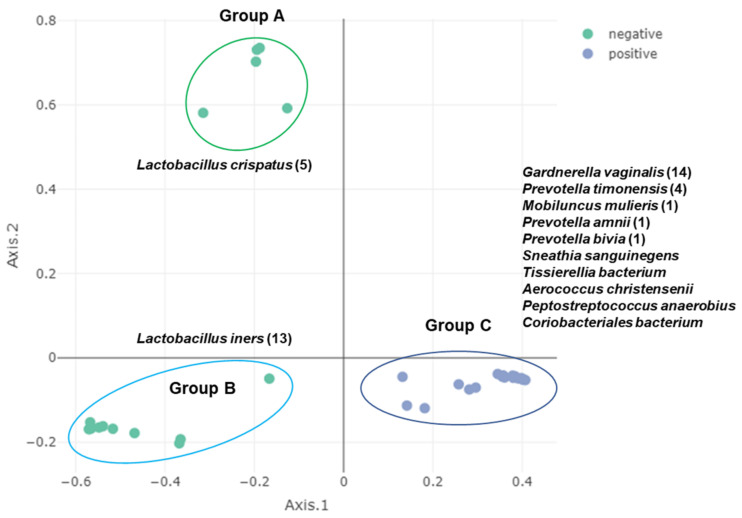
PCoA based on mapped reads on 176 bins. The names of major bacterial species in each group are listed with the number of samples indicated in parentheses.

**Figure 15 ijms-23-01621-f015:**
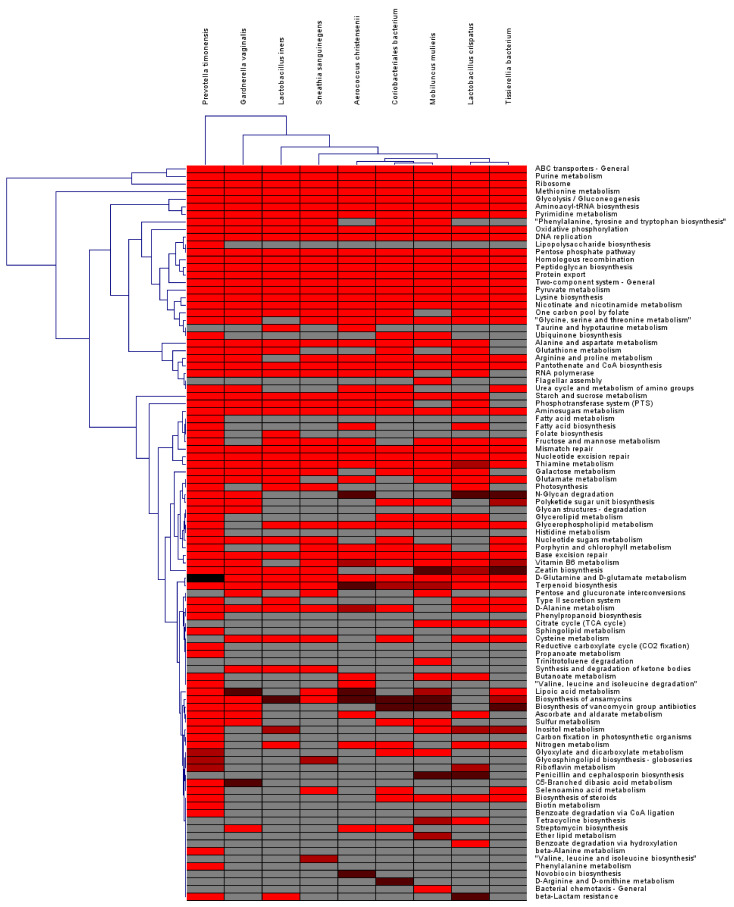
HCL of number of enzymes assigned to 97 KEGG pathways. As the number of enzymes increases, the red color becomes brighter. Gray color indicates no enzyme was identified in the given bacterial species.

**Table 1 ijms-23-01621-t001:** Summary of taxonomical classification of vaginal contigs.

Taxonomy	Number of Contigs	Proportion	No. of Taxonomies
Super kingdoms	276,994	96.80%	4
Phyla	261,853	91.50%	25
Classes	252,721	88.30%	32
Orders	242,180	84.60%	58
Families	218,460	76.30%	99
Genera	192,420	67.20%	209
Species	122,133	42.70%	339

## Data Availability

All datasets used in this study are accessible in the European Nucleotide Archive with accession number PRJEB21446.

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
