# Peer review of "De Novo Assembly and Annotation of the Vaginal Metatranscriptome Associated with Bacterial Vaginosis"

_ijms, 2022, doi:10.3390/ijms23031621_

Round 1
Reviewer 1 Report
Submitted paper is well written and provides essential information on not only vaginal microbiome but also on efficiency of different analysis approaches. These data would undoubtedly be helpful in further studies to implement proper investigation protocol in context of microbiome analysis. The authors should only verify quality of the figures included within the manuscript to make sure that best presentation would be obtained when published.
Author Response
Thank you so much for your kind comments. We have provided a high quality of figures with 600 dpi.
Reviewer 2 Report
The authors aimed to reanalyzed 40 vaginal transcriptomes from a previous study by de novo assembly followed by functional annotation using the Squeeze Meta analysis pipeline. Moreover, they evaluated the difference between the short read-based classifier and de novo assembler-based approach for the microbiome community study. Finally, they obtained 172 bins containing known and unknown bacterial genomes with functional classification of expressed genes.
The study covers some issues that have been overlooked in other similar topics. The structure of the manuscript appears adequate and well divided in the sections. Overall, the manuscript was written in good English and easy to understand and follow. Some of the comments that would improve the overall quality of the study are:
1-) Limitations and 2-) conclusions of the study are missing. Please add it.
Author Response
Comment 1: 1-) Limitations and 2-) conclusions of the study are missing. Please add it.
Response: We have already included limitations of our study in the discussion as follows. In addition, we have included a section for conclusions in the manuscript.
An NR protein database covers most known organisms, while the other two methods only include a limited number of bacteria for which genome sequences are available. Further-more, the approach based on BLASTX search using translated nucleotide sequences against NR sometimes leads to false positive taxonomy identification, although BLASTX search has very high gene prediction accuracy [41]. For example, complete genome sequences of an organism obtained using BLASTX search can be matched to many organisms that are closely related to the target organism.
The lack of bacterial genomes in the reference database could result in the misinterpretation of microbiome analyses. Therefore, it is very important to obtain as many refer-ence bacterial genomes as possible for microbiome studies.
Here, we obtained 176 metatranscriptome-assembled genomes (MAGs). As the MAGs were derived from the transcriptomes in this study, the completeness of MAGs was lower than that of metagenome-assembled genomes [25].
Therefore, it was very difficult for us to identify specific gene functions associated with BV in the up-regulated COGs since the number of identified COGs for the up-regulated group was much higher than that for the down-regulated group in most functional categories. However, the approach of dividing the COGs into 27 functional categories did narrow down the specific functions enriched in the up-regulated and down-regulated COGs.
KEGG analysis revealed that biofilm formation-associated genes for V. cholerae, E. coli, and P. aeruginosa were strongly up-regulated in BV-associated samples. However, those three bacterial species were not identified in our study. It is likely that many orthologous genes associated with biofilm formation are up-regulated in BV-associated samples.
- Conclusions
In this study, reanalysis using 40 vaginal transcriptome datasets based on de novo assembly and functional annotation was useful to define the vaginal microbiota commu-nity associated with BV. We found that metaT-Assembly identified a higher number of bacterial species than 16S rRNA and metaT-Kraken; however, metaT-Assembly and metaT-Kraken exhibited similar major bacterial composition at the species level. Binning of metatranscriptome data provided single transcriptomes of major known bacteria as well as several unidentified bacteria in the vagina. Functional analyses based on COGs and KEGG pathways suggested that a high number of transcripts were expressed by the microbiome complex in the BV-associated samples as compared to that of the non-BV-associated samples. The KEGG pathway analysis with an individual MAG iden-tified the specific functions of the identified individual bacterial genome. Taken together, we demonstrated that the metaT-Assembly approach is an efficient tool to understand the dynamic microbial communities and their functional roles associated with the human vagina.
